# Conserved visual capacity of rats under red light

Nader Nikbakht[1,2]*, Mathew E Diamond[1]*

[1]Tactile Perception and Learning Lab, International School for Advanced Studies (SISSA), Trieste, Italy; [2]Department of Brain and Cognitive Sciences, McGovern Institute for Brain Research, Massachusetts Institute of Technology, Cambridge, United States

**Abstract** Recent studies examine the behavioral capacities of rats and mice with and without visual input, and the neuronal mechanisms underlying such capacities. These animals are assumed to be functionally blind under red light, an assumption that might originate in the fact that they are dichromats who possess ultraviolet and green cones, but not red cones. But the inability to see red as a color does not necessarily rule out form vision based on red light absorption. We measured Long-Evans rats' capacity for visual form discrimination under red light of various wavelength bands. Upon viewing a black and white grating, they had to distinguish between two categories of orientation: horizontal and vertical. Psychometric curves plotting judged orientation versus angle demonstrate the conserved visual capacity of rats under red light. Investigations aiming to explore rodent physiological and behavioral functions in the absence of visual input should not assume red-light blindness.

*For correspondence:
nikbakht@mit.edu (NN);
diamond@sissa.it (MED)

Competing interests: The authors declare that no competing interests exist.

## Introduction

Rats, like many rodents, are largely crepuscular and, even during daylight, are usually to be found in poorly illuminated environments (*Macdonald et al., 1994*). Their retina is rod dominated, with cones making up as little as 1% of photoreceptors (*Jacobs et al., 2001*; *La Vail, 1976*). Rods are about 100 times more sensitive (*Yau, 1994*) and allow vision under low-light conditions, with a peak spectral absorption of rhodopsin in rods in rats and mice at 498 nm (*Bridges, 1959*; *Govardovskii et al., 2000*; *Lyubarsky et al., 2004*). Although rats are not color-blind (*Jacobs et al., 2001*; *Lemmon and Anderson, 1979*; *Muenzinger and Reynolds, 1936*; *Munn and Collins, 1936*; *Walton and Bornemeier, 1938*), they perform poorly in discriminating between nearby wavelengths, compared to humans (*Walton, 1933*). Rats' color sensitivity is based on two sets of cones with peak sensitivities in the range of ultraviolet (UV cones; peak absorption at 358–359 nm) and green (M cones; peak absorption at 509–510 nm) (*Deegan and Jacobs, 1993*; *Jacobs et al., 1991*; *Szél and Röhlich, 1992*). Recently, electroretinogram (ERG) responses of the photopic spectral sensitivity curves of photoreceptors of rats and mice were measured throughout the UV–visible spectrum (300–700 nm) (*Rocha et al., 2016*). These measurements identified two sensitivity peaks in Wistar rats: 362 and 502 nm; no significant response to long wavelength light (above 620 nm) was detected. This study reinforced the already existing notion that red light is experienced as a total absence of usable light. In contrast, another study (*Niklaus et al., 2020*) showed significant scotopic and photopic ERG responses to red light even at low intensities. In the present study, we challenge the notion of form vision blindness under red light and find, contrary to expectation, good behavioral performance.

## Results

Rats were required to categorize the orientation of a solid disk-like object with a circular boundary and raised parallel bars, alternately colored white and black, thus forming a square-wave grating (*Figure 1A*). Orientations in the range of 0°– 45° were rewarded as horizontal and orientations in the range of 45°– 90° as vertical (*Figure 1B*). *Figure 1C* illustrates the sequence of events in the behavioral task. Each trial started with the rat's head poke, which triggered the opening of an opaque gate, followed by illumination with light sources of various wavelengths. A transparent panel in front of the object prevented the rat from generating tactile cues. After observing the object, the rat turned its head toward one spout (L or R) and licked. The boundary angle, 45°, was rewarded randomly on left or right. Illumination was by a white LED array or else by monochrome LEDs with peak intensities at 626 nm, 652 nm, 729 nm, 854 nm, and 930 nm as measured by spectrometer and verified by the manufacturer's datasheet (*Figure 1D*; see Materials and methods). Half widths were 12.1–49.1 nm.

To quantify rats' performance, we used a cumulative Gaussian function to fit psychometric curves to the data of each rat (see Materials and methods). *Figure 1E* reveals that all rats (N = 4, pale curves; average curve in dark blue) performed well under white LED illumination.

Sessions with white light were interspersed with sessions illuminated by various narrow-band monochrome LEDs in the range of red, far-red, and infrared (*Figure 1D*). It is important to perform behavioral testing under illumination with narrow-band monochromatic light sources to eliminate the possibility of off-peak illumination through the tail of a wide power spectral distribution. *Figure 2A, B* shows that dark-adapted rats performed the visual categorization task under 626 nm and 652 nm LEDs (perceived as red by humans) with accuracy equivalent to that under white light, refuting the common belief of functional blindness under red light. Rats performed well even under peak 729 nm (*Figure 2C*), perceived as far-red by humans, though clearly diminished with respect to white. They performed poorly under infrared illumination, comprising 854 nm and 930 nm light (*Figure 2D,E*).

The performance (N=four rats) under different illumination wavelengths is summarized by the cumulative proportion of trials categorized correctly across all orientations (*Figure 2F*). Average performance was 87 ± 2% correct under white light (confidence interval is SEM across rats). Average performance was 84 ± 3% under 626 nm illumination (p=0.158 compared to white light; bootstrap tests illustrated in *Figure 2—figure supplement 1*), 86 ± 2% under 652 nm (p=0.419 compared to white light), 72 ± 3% under 729 nm (p=0.00 compared to white light), 53 ± 2% under 854 nm (p=0.257 compared to chance), and 50 ± 1% under 930 nm average performance (p=0.316 compared to chance).

Visual acuity under the red LEDs (626 nm and 652 nm) cannot be explained by unintended 'leakage' toward shorter wavelengths. *Figure 1D* shows that the infrared LEDs (854 nm and 930 nm) emitted ~2–50 times higher intensity at shorter wavelength than did the red LEDs, yet performance under illumination by IR LEDs was at chance. Therefore, performance under red LEDs is better explained by positing that enough red light was absorbed to guide the discrimination.

## Discussion

Rodent behavioral experiments done under red light are believed to place the animal in dark conditions while allowing the experimenter to observe the preparation directly or by video recording (*Celikel and Sakmann, 2007*; *Cloke et al., 2015*; *Diamond et al., 1999*; *Englund et al., 2020*; *van Goethem et al., 2012*; *Harris and Diamond, 2000*; *Harris et al., 1999*; *Jacklin et al., 2016*; *Nikbakht et al., 2012*; *Pacchiarini et al., 2020*; *Reid et al., 2014*; *Salaberry et al., 2017*; *Sieben et al., 2015*; *Vasconcelos et al., 2011*; *Winters and Reid, 2010*). The relevant literature includes place-cell studies of rats navigating under dim red light, taken as the absence of visual input (*O'Keefe, 1976*; *Save et al., 2000*; *Zhang and Manahan-Vaughan, 2015*; *Zhang et al., 2014*). In a developmental study of the visual cortex, mice were dark-reared but were allowed very brief daily exposure to red light (*Kowalewski et al., 2021*). Furthermore, red illumination is often used in reverse light cycle conditions in animal husbandry settings with the assumption that it will not affect rodent circadian rhythms (*Emmer et al., 2018*). In all of these cases, the animals may acquire more visual input than previously believed.

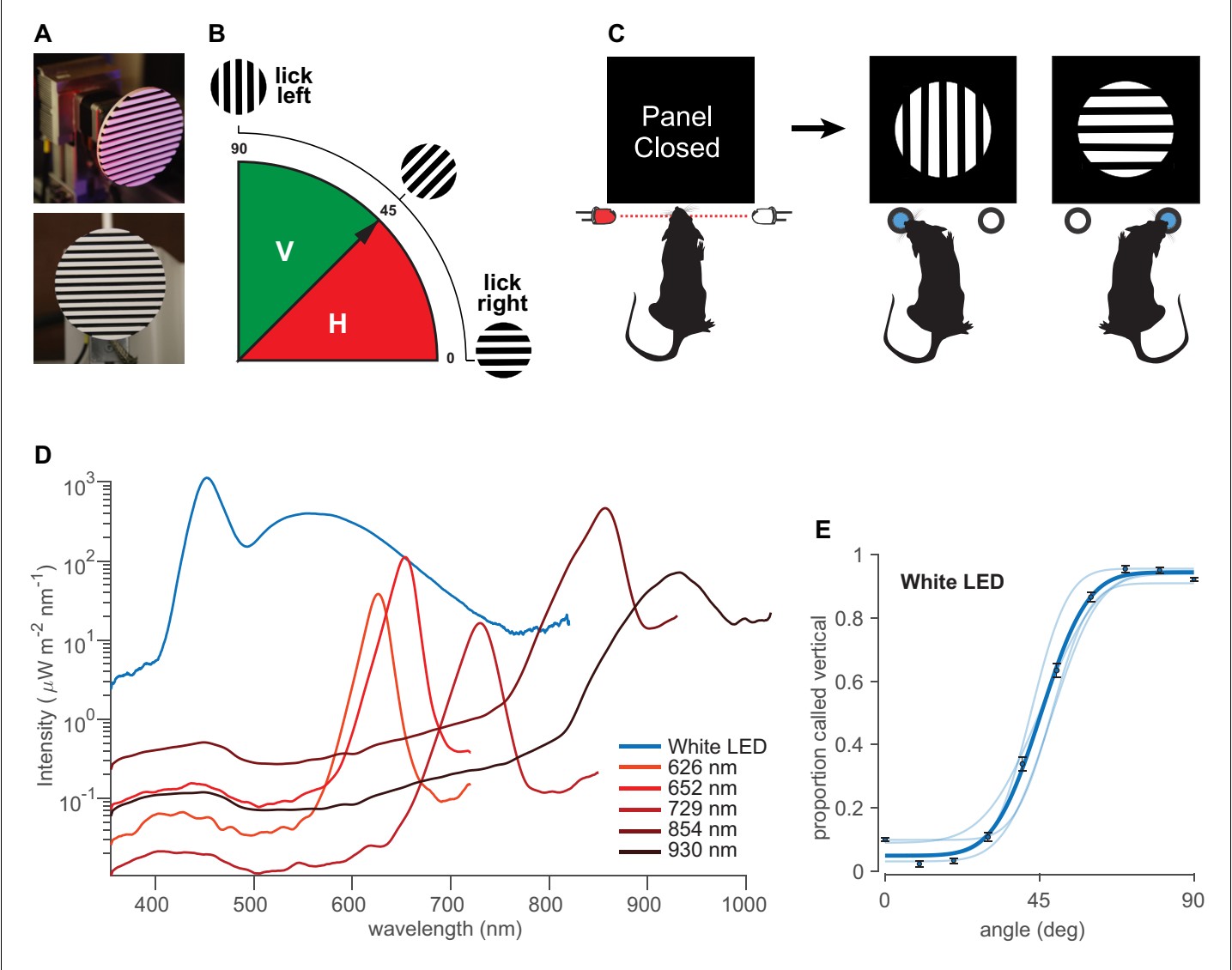

**Figure 1.** Orientation categorization task. (A) Discriminandum viewed at an oblique angle (top) and exactly from the front (bottom), the latter approximating the perspective of the rat. (B) Schematic of the orientations of the stimuli and rule of the categorization task. 0˚–45˚ (red) rewarded as 'horizontal' and 45˚–90˚ (solid green) rewarded as 'vertical.'. (C) Sequential steps in the behavioral task. Each trial started with a head poke that interrupted a light beam and triggered the opening of an opaque gate, followed by visual access to the object. After probing the stimulus, the rat turned its head toward one spout, in this illustration left for vertical and right for horizontal. See *Figure 1—figure supplement 1* for the experimental setup. (D) Irradiance for each LED, measured by a spectrometer at the stimulus delivery area. Equivalent photon count values are shown in *Figure 1—figure supplement 2*. Integrating the power under the curves at wavelengths <580 nm reveals that the infrared LEDs emitted from 2 to 50 times higher intensity as compared to red LEDs, in spite of the infrared LEDs being centered at longer wavelengths (infrared 854 nm and 930 nm LEDs emitted 1.95 and 0.48 mW/cm$^2$, respectively, while red 626, 652, and 729 nm LEDs emitted 0.12, 0.28, and 0.04 mW/cm$^2$, respectively). See *Figure 1—figure supplement 3* for normalized irradiance for each LED along with rat's photopigment spectral sensitivity. (E) Pale curves give the performance of four rats under white light. Dark data points and curves show the average over all rats. Error bars are 95% binomial confidence intervals. See *Figure 1—figure supplement 1A* for the experimental setup.

The online version of this article includes the following figure supplement(s) for figure 1:

**Figure supplement 1.** Details of the experimental setup.

**Figure supplement 2.** Log-scaled irradiance for each LED, measured by a spectrometer at the stimulus delivery area in units of *photons.μm$^{-2}$.s$^{-1}$.nm$^{-1}$*.

**Figure supplement 3.** Log-scaled normalized irradiance for each LED along with a template that approximates the photopigment spectral sensitivity of rat's rhodopsin as well as S-opsin, M-opsin (*Stockman and Sharpe, 2000*).

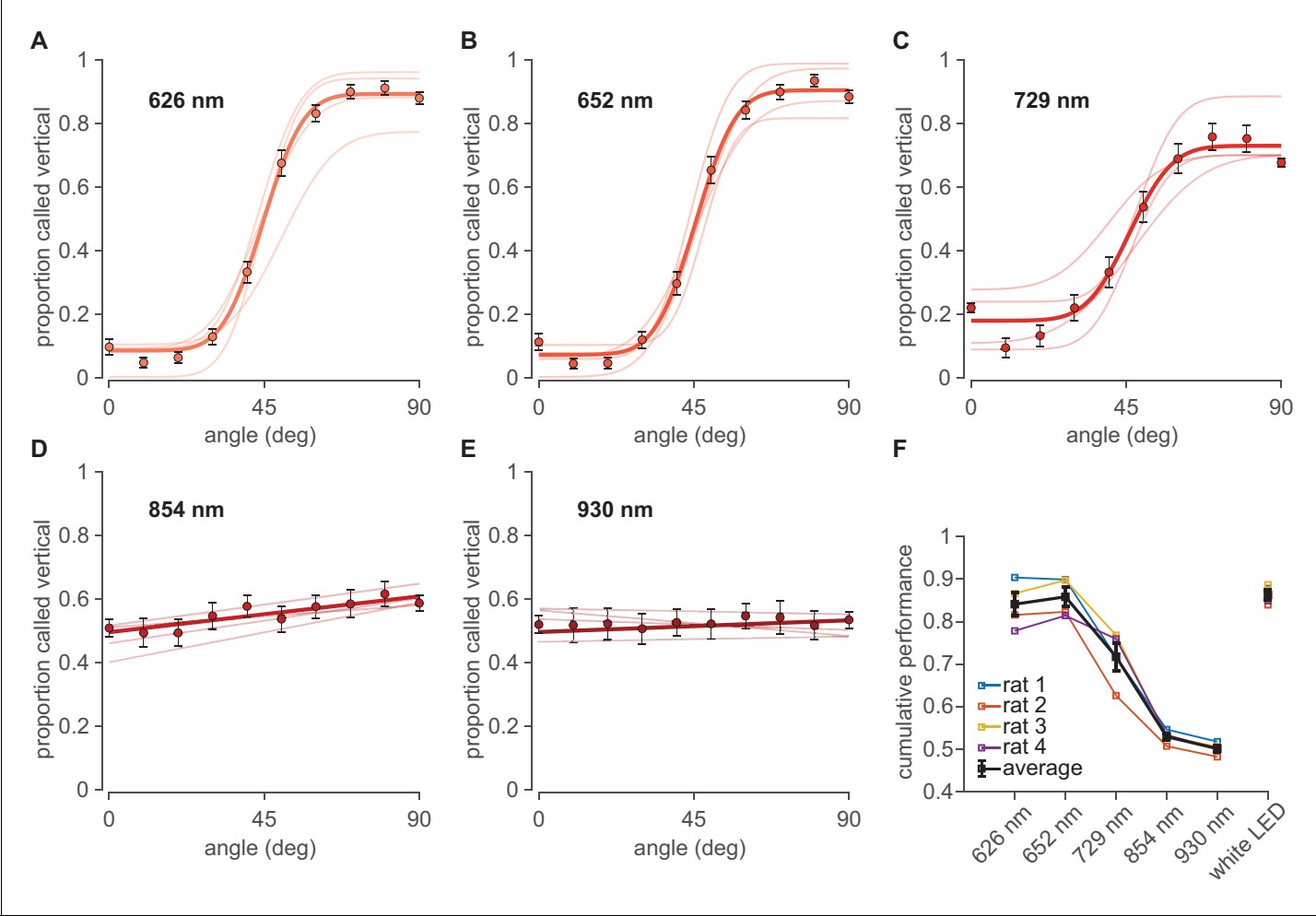

**Figure 2.** Performance under far-red and infrared illumination. (A–E) Psychometric curves obtained under illumination with monochrome LEDs in the range of far-red to infrared with peak wavelengths at 626 nm, 652 nm, 729 nm, 854 nm and 930 nm, respectively. Pale curves depict the performance of four rats. Dark data points and lines or curves show the average over all rats. Error bars are 95% binomial confidence intervals. (F) Summary cumulative performance (proportion correct) for all four rats (colored) and the average rat (black) in each illumination condition. Data from all angles (except 45°, uninformative about discriminative capacity) are pooled. See *Figure 2—figure supplement 1* for tests of significance.

The online version of this article includes the following figure supplement(s) for figure 2:

**Figure supplement 1.** Details of the statistical test of significance.

**Figure supplement 2.** An example sequence of sessions.

Rodents lack red cones (***Deegan and Jacobs, 1993***; ***Jacobs et al., 1991***; ***Szél and Röhlich, 1992***), but from the inability to see red as a color, it does not necessarily follow that they cannot absorb red light through their rod-dominated retina to support form vision. A recent study (***Niklaus et al., 2020***) examined the retinal responses mediated by rods and cones of pigmented (Brown Norway) and albino (Wistar) rats in response to monochromatic far-red light of 656 ± 10 nm in both photopic (light-adapted) and scotopic (dark-adapted) settings. Both rat strains showed significant scotopic and photopic ERG responses to red light even at low intensities. These results hinted that even far-red light may provide effective illumination for rats. However, whether the photoreceptor activation by red light can lead to functionally meaningful signals has not yet been established. Excitation of retinal receptors by red light does not, by itself, indicate behavioral availability of the signal. For instance, red light might act to entrain circadian rhythms, unconscious to the animal. The sensitivity of mice is emphasized by the finding that red illumination delivered into the brain in an optogenetic protocol evoked a behavioral artifact in mice performing a visually guided discrimination task (***Danskin et al., 2015***).

Although our study does not include direct measures of photoreceptor sensitivity, the results indicate that mechanisms must exist for converting miniscule quantities of receptor activation into meaningful signals, perhaps through some cross-retina population code. It is interesting to note the possibility that photoreceptors could respond to longer wavelengths through a nonlinear optical process including two-photon activation of photopigments (*Palczewska et al., 2014*; *Vinberg et al., 2019*), although the emission intensities used in our work are many orders of magnitude lower than those used in two-photon studies (*Palczewska et al., 2014*).

In the present work, Long-Evans rats demonstrated substantial visual form capacity under illumination by LEDs emitting long-wavelength red and far-red light – LEDs that are low cost, easy to use, and thus common in the laboratory. Object discrimination began to degrade when illumination wavelength increased from 652 to 729 nm and was almost nil at 854 nm. The performance curves suggest the optimal conditions for vision-excluded behavioral studies: illumination at 850 nm did not support visual form capacity, yet remains within the sensitivity range of inexpensive silicone detectors (CMOS and CCDs). Thus, behaviors may be documented by video while the animal performs without visual cues.

Rats and mice are the most frequently used laboratory mammals, ideal for research on spatial navigation (*Frank et al., 2000*; *MacDonald et al., 2011*; *Moser et al., 2017*; *O'keefe and Nadel, 1978*; *Pastalkova et al., 2008*; *Wood et al., 2000*) and the processing of tactile (*Diamond et al., 2008*; *Fassihi et al., 2017*; *Zuo and Diamond, 2019*) and olfactory (*Chae et al., 2019*; *Koldaeva et al., 2019*; *Uchida and Mainen, 2003*; *Uchida et al., 2006*) information. Neuroscientists have not traditionally attributed to rodents the wide range of visual perceptual functions characteristic of primates. However, there is growing interest in the use of rodents for the study of vision, alone or combined with other modalities (*Gharaei et al., 2018*; *Nikbakht et al., 2018*; *Nikbakht Nasrabadi, 2015*; *Sieben et al., 2015*; *Zoccolan, 2015*). Rats under broad-wavelength conditions spontaneously recognize an object even when views differ by angle, size, and position *Zoccolan, 2015*; such generalization is a hallmark of authentic visual perception and was once believed to belong only to primates. Importantly, they achieve high-level sensory-perceptual cognition through the workings of neuronal circuits that are accessible (*Crochet et al., 2019*; *Matteucci and Zoccolan, 2020*; *Matteucci et al., 2019*; *Steinmetz et al., 2019*; *Tafazoli et al., 2017*). The present study extends the range of visual perceptual functions for which rats can serve as models, characterizing their performance in judging object orientation and indicating the longest illumination wavelengths at which this capacity remains intact. A complete understanding of the visual processing of these animals is important not only in the design and control of the behavioral and physiological experiments but also to ensure optimal environmental lighting conditions for their well-being in laboratory settings.

## Materials and methods

### Experimental subject details

Four male Long–Evans rats (Charles River Laboratories, Calco, Italy) were used. They were caged in pairs and maintained on a 12/12 hr light/dark cycle; experiments were conducted during the light phase. Upon arrival, they were 8 weeks old, weighing approximately 250 g, and typically grew to over 600 g over the course of the study. They had free access to food in the cage. To promote motivation in the behavioral task, rats were water-deprived on days of training/testing. During each session, they received 17–22 mL of pear juice diluted in water (one unit juice: four units water) as reward. After the session, they were given access to water ad libitum for 1 hr, though they were rarely thirsty and then they were placed for several hours in a large, multistory-enriched environment to interact with other rats. Animals were examined weekly by a veterinarian. Protocols conformed to international norms and were approved by the Ethics Committee of SISSA and by the Italian Health Ministry (license numbers 569/2015-PR and 570/2015-PR).

### Behavioral method details

#### Apparatus

The main chamber of the apparatus, custom-built in opaque white Plexiglas, measured 25 × 25 × 37 (H × W × L, cm) (*Figure 1—figure supplement 1*). The rat started a trial by interrupting an infrared

beam detected by a phototransistor (*Figure 1C*). Beam interruption triggered fast opening of an opaque panel (through a rotational motion of 40° in 75 ms), actuated by a stepper motor, uncovering a circular hole (diameter 5 cm) in the front wall through which the rat could extend its head to see the object. The stimulus was 3 cm behind the opaque panel (further details below), and the reward spouts were 2 cm lateral to the edge of the stimulus. A transparent panel prevented direct touch.

The apparatus was in a Faraday cage that, with the door closed, provided acoustic, visual, and electromagnetic isolation. An array of 12 infrared emitters (λ = 930 nm, OSRAM Opto Semiconductors GmbH, Germany) illuminated the stimulus port to permit the investigator to monitor behavior and to execute video recording. Such illumination did not provide visual cues for the rat (see Results). For visual testing, different light sources were used to illuminate the stimulus: a pair of six white LED arrays or various high-power monochrome LED arrays (Roithner LaserTechnik, GmbH) with peak intensities at 626 nm (LED620-66-60), 652 nm (LED660-66-60), 729 nm (LED735-66-60), 854 nm (LED850-66-60), and 930 nm (LED940-66-60) (see *Figure 1D* for Gaussian fits to the measured spectrum in ambient temperature). The power spectral distributions were measured with a calibrated spectrometer and the associated software (Model: FLAME-S-XR1-ES, OceanOptics, Rochester, NY) via a visible-NIR fiber (core diameter 200 μm) attached to a cosine corrector. The cosine corrector acts as an optical diffuser that couples to the optical fiber and spectrometer to collect signals from 180° field of view. The spectrometer was calibrated using a Deuterium-Halogen calibration light source unit (DH-3 plus OceanOptics, Rochester, NY). This light source provides known absolute intensity values at several wavelengths, expressed in μW/cm$^2$/nm. The spectral intensities emitted by this source are calibrated according to the intensity standard of the National Institute of Standards and Technology (NIST). The spectrometer calibration procedure accounts for the different sensitivities of the spectrometer across different wavelengths. The spectrometer we used has a software suite that reports the values measured from the calibrated spectrometer as spectral irradiance (the power received by a surface per unit area of the measurement probe per wavelength [in units of μW/cm$^2$/nm]).

Two infrared-sensitive video systems (Point Grey Flea, Edmund Optics, Barrington, NJ) registered the rat's actions. The first camera, equipped with a macro lens (Fujinon TV HF25HA-1B Lens, Fujifilm, Tokyo) mounted 25 cm above the stimulus delivery area (distance with respect to the center of stimulus), monitored the rat's interaction with the object. In some sessions, this camera was set to 250 f/s to monitor head, snout, and whisker position and movement during behavior. The second camera provided a wide-angle view (Fujinon HF9HA-1B Lens, Fujifilm, Tokyo) and monitored the entire setup, illuminated with adjustable infrared LEDs, at 30 f/s.

Reward spouts included custom-made infrared diode sensors interrupted by the tongue. Only the licking signal from the correct spout triggered the pump motor (NE-500 programmable OEM; New Era Pump Systems, mounted on a vibration-cancellation pedestal) to extrude the reward, 0.05 mL per trial of diluted pear juice. Licking marked the end of the trial, accompanied by the closure of the opaque front panel. Before the next trial began, the motor on which the stimulus was mounted rotated to generate the next orientation.

Custom-made software was developed using LabVIEW (National Instruments, Austin, TX). An AVR32 board (National Instruments) and multiple Arduino Shields (National Instruments) acquired all sensor signals and controlled the motors, LEDs, and the reward syringe pumps. All the sensors, actuators (including motors and pumps), and lights were interfaced with the computer program allowing full control over a wide range of parameters governing the flow of the training and testing. Although fully automatic, the software allowed the experimenter to modify all the parameters of the task and control the lights, sensors, and motors online as needed.

## Visual stimulus presentation

The stimulus was a black and white square-wave grating within a circular 9.8 cm diameter circumference, built in-house by a 3D printer (3D Touch, BFB Technologies, *Figure 1A*). It was mounted on a stepper motor and rotated to generate the trial's intended orientation (*Figure 1B*). The stimulus stepper motor was controlled through a feedback system with a digital step counter to maintain the exact desired orientation. In the study that originally explored orientation judgment (*Nikbakht et al., 2018*), tactile exploration of the object was allowed in some trials, but the tactile condition is not considered in the present work; only visual, touch-free data are included. Within

behavioral testing sessions, each stimulus orientation was sampled from a uniform distribution in 5° steps between −45° and 135° and presented in a semi-random fashion (sampling without replacement). For analysis, we binned the angles every 10°. Visual acuity is measured in cycles per degree (cpd), an assessment of the number of lines that can be seen as distinct within a degree of the visual field:

$$Acuity = \frac{1}{2 \times tan^{-1}\left(\frac{h}{d}\right)} cpd$$

where $h$ is the width of each line in the stimulus and $d$ is the distance from the eye. Considering the 3 cm distance behind the opaque panel (*Figure 1—figure supplement 1*), at the moment of panel opening, each cycle of the 14 cycles would occupy $2tan-1(3.5/30) = 13.31°$ of visual angle, for a total stimulus coverage of about 117°. The spatial frequency of the gratings would be $1/(2tan-1(3.5/30)) = 0.075$ cycles per degree of visual angle. As the normal visual acuity of Long–Evans rats has been estimated as ~1 cpd (*Prusky et al., 2002*), the bars would be expected to be resolvable. Rats also have a large depth of focus, from 7 cm to infinity (*Powers and Green, 1978*). The width of the binocular field directly in front of the rat's nose, generally considered the animal's binocular viewing area (*Mei et al., 2012*), ranges from approximately 40°–110°, depending on head pitch (*Wallace et al., 2013*). The 117° stimulus should thus completely cover the rat's binocular visual field.

When all illumination sources were turned off, the ambient light magnitude was 0 cd/mm$^2$ (Konica Minolta LS-100 luminance meter, Tokyo).

## Quantification of LED emission characteristics

Following the procedure of *Franke et al., 2019*, for each LED, we computed the following measurements, which we have included in Supplementary table 2.

Using the spectrophotometer measurements ($P(\lambda)[\mu W.cm^{-2}.nm^{-1}]$), we can calculate the spectral power density for each LED:

$$I_{LED}(\lambda) = 10^{-2} \times \int_{\lambda_{min}}^{\lambda_{max}} P(\lambda)d(\lambda)\left[W.m^{-2}.nm^{-1}\right]$$

The energy of a single photon at wavelength $\lambda$ is $Q_p(\lambda) = h \cdot c/\lambda [m^2.kg.s^{-1}]$, where $c = 299,792,458 m.s^{-1}$ is the speed of light, and $h = 6.62607004 \times 10^{-34}[J.s]$ is Planck's constant.

Then we computed the spectral photon count for each LED ($SPC(\lambda)[photons.\mu m^{-2}.s^{-1}.nm^{-1}]$):

$$SPC(\lambda) = \frac{I_{LED}(\lambda)}{Q_p(\lambda)} = 10^{-12}/(h \times c) \times I_{LED}(\lambda) \times \lambda \left[photons.\mu m^{-2}.s^{-1}.nm^{-1}\right]$$

Given the $d = 50 cm$ distance between the LEDs and the stimulus delivery area, the rat's dilated pupil area of $A_{pupil} = 0.0314 cm^2$, the rat's retinal surface area of $A_{retina} = 0.8 cm^2$, and the rat's rod outer segment area of $A_{rod} = 4 \times 10^{-8} cm^2$ (*Mayhew and Astle, 1997*), we approximated the photon flux density (C) of an isotropic LED radiation pattern at each photoreceptor as:

$$C = \frac{2.A_{pupil}.A_{rod}}{4\pi d^2.A_{retina}} \int_{\lambda_{min}}^{\lambda_{max}} SPC(\lambda)d(\lambda)\left[photons.\mu m^{-2}.s^{-1}\right]$$

Then, the photon flux per photoreceptor is:

$$C_{flux} = 10^8.C.A_{rod}\left[photons.s^{-1}\right]$$

## Behavioral task and training

Duration of training to reach stable performance was typically 4–6 weeks, with one session per day, and varied according to individual differences in rate of learning. The sequence of sessions for each rat was randomized, with each day's session illuminated by either red, infrared, or white LEDs, with equal likelihood. The standard training was done with stimulus illumination by white LEDs. The training protocol proceeded across a sequence of stages given in *Nikbakht et al., 2018*. Exclusion of unintended stimulus cues (olfaction, rotating motor noise) was controlled as given in

*Nikbakht et al., 2018*. Rats were dark-adapted in a light-free environment for 20–30 min prior to each session.

## Quantification and statistical analysis

### Analysis of behavioral data

We analyzed the behavioral data in MATLAB (MathWorks, Natick, MA) and LabVIEW. To quantify a single rat's performance, we fit psychometric curves to its choice data. For a given orientation, we calculated the proportion of trials categorized as vertical. Ideally, rats would categorize all trials with angle greater than 45° as vertical and all trials with angle less than 45° as horizontal. For 45° trials, choices should be evenly distributed between vertical and horizontal. However, task difficulty grows in the vicinity of 45°, such that real performance is better described by a sigmoid function with an inflection point at the point of subjective equality (PSE), the orientation at which subjects report the stimulus with equal likelihood as horizontal or vertical. In unbiased rats, the PSE should be at 45°. We generated psychometric functions using a cumulative Gaussian function with the general form given in the equation below based on (*Wichmann and Hill, 2001*). The parameter estimation was then performed in MATLAB using maximum-likelihood estimation:

$$\psi(x; \mu, \sigma, \gamma, \lambda) = \gamma + (1 - \gamma - \lambda)F(x; \mu, \sigma).$$

The two-parameter function, $F(x; \mu, \sigma)$, is defined by a cumulative Gaussian distribution, as follows:

$$F(x; \mu, \sigma) = \frac{1}{2}\left[1 + erf\left(\frac{x - \mu}{\sigma\sqrt{2}}\right)\right]$$

where $x$ is the stimulus orientation, $\gamma$ is the lower bound of the function $\psi$, and $\lambda$ is the lapse rate. Often, $\gamma$ and $\lambda$ are considered to arise from stimulus-independent mechanisms of guessing and lapsing. $\mu$ is the mean of the probability distribution that determines the displacement along the abscissa of the psychometric function – a reflection of the subject's bias – and $\sigma$ is the standard deviation of the cumulative Gaussian distribution. $\sigma$ determines the slope of the psychometric function, a common measure of acuity.

Where fitting with a sigmoid was not appropriate (*Figure 2D,E*), data were fit with the line $y = ax + y_0$, where $x$ is the stimulus orientation, $a$ is the slope, and $y_0$ is the y-intercept.

### Test of significance for the fitted psychometric curves

Errors around the performance value for each orientation and modality condition were expressed as a 95% binomial proportion confidence interval computed by approximating the distribution of errors about a binomially distributed observation, $p$, with a normal distribution:

$$\hat{p} \pm 1.96\sqrt{\frac{1}{n}\hat{p}(1 - \hat{p})}$$

where $p$ is the proportion of correct trials (Bernoulli) and $n$ is the number of trials.

For statistical tests of significance, we performed a non-parametric test based on bootstrapping, as follows. We computed a distribution of the performance values from the fitted psychometric functions based on 1000 resamples of the behavioral data. We then performed pairwise comparisons between all the performance values generated via bootstrapping from fitted psychometric functions of each experimental condition, calculated the overlap between the distributions and computed the p-values.

## Acknowledgements

We acknowledge the financial support of the Human Frontier Science Program (MED) (http://www.hfsp.org; project RGP0015/2013) and the European Research Council advanced grant CONCEPT (MED) (http://erc.europa.eu; project 294498). The Regional Laboratory for Advanced Mechatronics, LAMA FVG (http://lamafvg.it), supported the design and construction of custom instrumentation. The funders had no role in study design, data collection and analysis, decision to publish, or

preparation of the manuscript. Erik Zorzin assisted in spectral measurements. We are grateful to members of the Diamond lab for fruitful comments and discussions.

## Additional information

### Funding

| Funder | Grant reference number | Author |
| --- | --- | --- |
| European Research Council | 294498 | Mathew E Diamond |
| Human Frontier Science Program | RGP0015/2013 | Mathew E Diamond |

The funders had no role in study design, data collection and interpretation, or the decision to submit the work for publication.

### Author contributions
Nader Nikbakht, Mathew E Diamond, Conceptualization, Supervision, Funding acquisition, Resources, Data curation, Software, Formal analysis, Validation, Investigation, Visualization, Methodology, Writing - original draft, Writing - review and editing

### Author ORCIDs
Nader Nikbakht ⓘD https://orcid.org/0000-0002-9726-9115
Mathew E Diamond ⓘD https://orcid.org/0000-0003-2286-4566

### Ethics
Animal experimentation: The rats were under the care of a consulting veterinarian. Study protocols conformed to international norms and were approved by the Ethics Committee of SISSA and by the Italian Health Ministry (license numbers 569/2015-PR and 570/2015-PR).

### Decision letter and Author response
Decision letter https://doi.org/10.7554/eLife.66429.sa1
Author response https://doi.org/10.7554/eLife.66429.sa2

## Additional files

### Supplementary files
• Supplementary file 1. LED emission characteristics. Measured intensity of each LED in mW/cm$^2$ and total irradiated power (in mW), as measured through our spectrometer (Supplementary table 1) and Additional LED emission characteristics (Supplementary table 2).

• Transparent reporting form

### Data availability
All data generated or analyzed during this study will be included in the manuscript and supporting files. Source code files will be provided for Figures 1 and 2 at https://github.com/nadernik/nikbakht_diamond_elife (copy archived at https://archive.softwareheritage.org/swh:1:rev:4ba735b86cbcb4bb0094d1d8531a36297561752a).

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
