## [Decision Letter]

**Acceptance summary:**

It is commonly thought that rodents are functionally blind when their surroundings are illuminated with light of longer wavelengths, which humans perceive as red. Nikbakht and Diamond challenge this assumption, and show that rats can accurately discriminate different objects that are illuminated only by red light. This result has important implications for the design of experiments and housing of rodents, and demonstrates that rodents can perform perceptual tasks despite the weak activation of retinal photoreceptors.

**Decision letter after peer review:**

Thank you for submitting your article "Conserved visual capacity of rats under red light" for consideration by *eLife*. Your article has been reviewed by 4 peer reviewers, one of whom is a member of our Board of Reviewing Editors, and the evaluation has been overseen by Joshua Gold as the Senior Editor. The following individuals involved in review of your submission have agreed to reveal their identity: Nick Steinmetz (Reviewer #3); Katrin Franke (Reviewer #5).

The reviewers have discussed their reviews with one another, and the Reviewing Editor has drafted this to help you prepare a revised submission. Overall the reviewers were very positive about this study. They identified a number of Discussion points and further analyses to improve the paper.

Essential revisions:

1) We recommend that the authors comment on at least three applications that they haven't mentioned, which are additional points of impact of the study:

i) Studies of visual development and plasticity in rodents often use "dark-rearing" or days/weeks of darkness as a way to test the visual system under conditions of no stimulation; however, red light is used in husbandry tasks during these times, and any visual experience during those times could severely confound the interpretation of such studies (to pick one recent example, Kowalewski.… Kuhlman Current Biology 2020).

ii) Red light sources are often used in optogenetic perturbation experiments and might be assumed to be invisible to the subject, an assumption clearly invalid. In fact there was a paper with a similar finding to the authors (and to Niklaus et al) in this context – Danskin… Waters Plos One 2015.

iii) Quite some studies nowadays do eye or whisker monitoring and some conclusions of neural activity in the dark could very well be affected by these conclusions. Another point is place field monitoring in the dark – an experiment that has been quite fundamental for the hippocampal field.

2) The photon absorption rate depends on the sensitivity curve of the visual pigment expressed by the photoreceptor and on the light intensity. As stated by the authors in the Discussion, it has been shown that mammalian photoreceptors can be activated by infrared light if the intensity is strong enough – both by single photon and two photon events. As a result of this dependence on intensity, it is critical that the authors carefully measure and report the light intensity used in their experiments and ideally relate this to the expected photoreceptor activation, e.g. as photo-isomerizations per second. In addition, we suggest to discuss the rationale behind using this intensity range in their experiments. Together, this will help other researchers to interpret these interesting behavioral results and adjust their experimental setup accordingly.

The authors did not attempt to present a matching white light that would activate the rods or the cones in the same manner and investigate adaptation. This should be discussed and considered. Furthermore analysis on the extent to which the cones and the rods would be driven in the red light range would be very helpful.

The way in which the LED data is presented and analyzed should be improved. In particular, the authors seem to show fits and they are shown on a linear scale. The authors should show the raw spectrometer values and show these on a logarithmic scale in proper physical units in order to assess how much energy the LED generates in the normal vision range. With respect to intensity calibration of light sources, we suggest to include additional information in the manuscript. The authors mention in the Methods that they use a calibrated spectrometer for intensity measurements. Does this also account for different sensitivity of the spectrometer across different wavelengths? How do you transform spectrometer output to irradiances shown in Figure 1—figure supplement 2? And what is the unit of irradiance in this figure? Did the authors calibrate the different LEDs such that they result in the same irradiance during experiments? We suggest to include all these points into the Methods section. Also, please use absolute and not normalized spectra to compare intensities across LEDs – see Figure 1D. In addition, we believe it is important to (i) state the absolute intensities used in the experiments in the Results section, (ii) relate these intensities to expected photoreceptor activation and also intensities used in common experimental designs, and (iii) discuss the rationale behind using these intensities during experiments. This will help the reader to interpret the results. For example, did the authors pick these intensities because they are regularly used to dark adapt rats during experiments? Did the authors also test other intensities?

If there would be any concern left here after these analysis, one solution could be to use monochromatic lasers for the experiment.

We further suggest to add a simple control experiment to exclude any light contamination in the experimental setup. For that, the authors should position the spectrometer head at the position of the rat's head during behavioral experiments and record both background light levels and spectra of all LEDs. While this is unlikely to interfere with the experiments, it is a fast test that could be added as a supplementary figure.

3) The paper needs to be improved in terms of considering the mechanisms:

i) The discussion on this needs to be improved and extended. These findings likely have a fairly straightforward explanation.

ii) It goes beyond the scope of the paper to investigate whether this depends on cones, rods or even two-photon effects (note that this is unlikely to explain the data given the wavelength results that they describe, this may have been a bit misplaced). The most straightforward explanation is that the small drive of the cones and rods, combined with adaptation mechanisms, is sufficient to perform the task at hand. This seems testable by using very low-intensity green light (hitting the M-cones and the rods) that elicits a similar drive. This would show there is nothing special about red here, it just means that rats can do this kind of task under very low-level lighting conditions.

iii) Another very simple experiment that could have been considered is to prevent adaptation, so first prevent adaptation by using green or white light at normal intensity and then put the animal in the red condition – this should prevent the animal from doing the task. The authors write that "Rats were dark-adapted in a light free environment for 20-30 minutes prior to each session". Was this a necessary step?

4) The normalized irradiance plot in 1D does not extend below 350nm though it is stated that rats have cones with peak sensitivity ~360nm. So, is it possible that the 630-730nm wavelength LEDs also give off a small amount of 300-350nm UV light, and this is used by the rats for discrimination? How can this be ruled out? Many long-pass optical filters also pass some light of much shorter wavelengths than the cutoff - to pick one example, see the spectrum for the DMLP650 on this page (https://www.thorlabs.com/newgrouppage9.cfm?objectgroup_id=3313). I don't know whether the LEDs in this study obtain their wavelength selectivity from anything at all related to the mechanisms of filtering in DMLP650 or similar - I'm not an expert - but it seems like it must be ruled out.

5) What accounts for the difference in findings relative to Jacobs et al 1991? I think this paper is the most classic/foundational, and it included a behavioral test of visual function, but concluded no behavioral sensitivity beyond 600 nm. I think the authors ought to indicate what they think accounts for the discrepancy in conclusions between that work and the present. Difference in species? Difference in behavioral task design? Difference in light source? Difference in range of wavelengths tested? Different analytical methods used to draw conclusions?

6) In the introduction, it is stated that Rocha et al 2016 found no ERG response above 620nm, while in the discussion it is stated that Niklaus et al 2020 found ERG responses to 650nm. Are the authors claiming that Rocha et al 2016 are simply wrong? If so, it would seem more appropriate to raise that study and call it into question in the discussion rather than the introduction - as is, that study is presented as accepted fact and is never revisited or questioned, which I found very confusing.

*Reviewer #1:*

This paper challenges a long-standing dogma in the rodent field, namely that rodents don't see in red light. The paper performs careful behavioral analyses to show that rats can indeed see (discriminate orientations) under red light condition. Even though the paper does not explain why this happens, the findings are likely of great interest to rodent vision scientists, and rodent researchers using infrared for e.g. housing and monitoring of the animal (eye, location). I have some concerns w.r.t. the purity of the LEDs. The discussion of the paper in terms of mechanisms should be strongly enhanced. I was surprised that the authors did not attempt to present a matching white light that would activate the rods or the M-cone in the same manner and investigate adaptation. Furthermore I expect more analysis on the extent to which the M cones and the rods would be driven in the red light range.

– The way in which the LED data is presented and analyzed should be improved. In particular, the authors seem to show fits and they are shown on a linear scale. The authors should show the raw spectrometer values and show these on a logarithmic scale in proper physical units in order to assess how much energy the LED generates in the normal vision range.

– Related to this, it would be straightforward to calculate how much the rods and cones are driven by the remaining white light and compare this to the extent to which they are driven by the red light.

– If there would be any concern left here after this analysis, I would recommend using monochromatic lasers for the experiment.

– The paper does little in terms of dissecting mechanisms:

1) The discussion on this needs to be improved and extended massively. I don't think these findings are mysterious but likely have a fairly straightforward explanation.

2) I think it goes beyond the scope of the paper to investigate whether this depends on cones, rods or even two-photon effects (note that this is unlikely to explain the data given the wavelength results that they describe, I felt this was a bit misplaced). it would be straightforward to obtain some minimum mechanistic insight. The most straightforward explanation is that the small drive of the cones and rods, combined with adaptation mechanisms, is sufficient to perform the task at hand. This seems testable by using very low-intensity green light (hitting the M-cones and the rods) that elicits a similar drive. This would show there's nothing special about red here, it just means that rats can do this kind of task under very low-level lighting conditions.

3) Another very simple experiment that could be added is to prevent adaptation, so first prevent adaptation by using green or white light at normal intensity and then put the animal in the red condition – this should prevent the animal from doing the task. The authors write that "Rats were dark-adapted in a lightfree environment for 20-30 minutes prior to each session". I guess, this was a necessary step.

– It would be useful to discuss a bit more practical implications. Quite some studies nowadays do eye or whiskerre monitoring and some conclusions of neural activity in the dark could very well be affected by these conclusions. I'm also thinking of place field monitoring in the dark – an experiment that has been quite fundamental for the hippocampal field.

*Reviewer #3:*

I think the authors actually undersell the full impact of this work, as it bears on at least two applications that they haven't mentioned:

1) Studies of visual development and plasticity in rodents often use "dark-rearing" or days/weeks of darkness as a way to test the visual system under conditions of no stimulation; however, red light is used in husbandry tasks during these times, and any visual experience during those times could severely confound the interpretation of such studies (to pick one recent example, Kowalewski.… Kuhlman Current Biology 2020);

2) Red light sources are often used in optogenetic perturbation experiments and might be assumed to be invisible to the subject, an assumption clearly invalid. In fact there was a paper with a similar finding to the authors (and to Niklaus et al) in this context – Danskin… Waters Plos One 2015.

*Reviewer #4:*

This study demonstrates that Long-Evans rats accurately discriminate the orientation of visual gratings illuminated by red light and far red light (wavelengths between 626 and 729), despite lacking a cone type with a peak absorbance at long wavelengths (>620). This important study follows recent work demonstrating that there are significant retinal responses to the same wavelengths of light in both rat and mouse species commonly used in behavioral and neural circuit function studies. The demonstration that rats retain clear form vision and can demonstrates accurate visually evoked behavior under "red-light conditions has important implications for studies of rodent visual perception which may have previously assumed "blindness" when the environment is illuminated by long wavelengths of light. Likewise, housing/rearing recommendations for rodents and possibly "dark" rearing studies not completed without any light, may be impacted by this study if experimenters used red light conditions to provide intermittent care for animals and assumed it was similar to no light or night-time conditions.

The major claims are well supported by the presented psychometric functions of the study subjects. The Authors also provide sufficient evidence that they accurately measured the intensity and wavelength span of their light sources and can therefore eliminate differences in light intensity, or overlap from neighboring wavelengths, as alternative explanations for their robust behavioral data.

This study will set a new standard and raise awareness of the need to stop considering "red-light" conditions as the same as no light or infrared conditions.

I'm at a loss to think of anything to improve the study substantively. It's a clear and tight study. The stimuli appear accurately measured and controlled- the behavior is beautiful (as much as behavior can be) and the specific study design was perfect for the question. It is clear these rats can see the stimuli and there doesn't appear to be any other explanation other than the ability of the rats to see in those narrow bands of red light.

*Reviewer #5:*

In the study "Conserved visual capacity of rats under red light", Nader Nikbakht and Mathew Diamond quantify the behavioral visual performance of rats under red and far-red light illumination. Due to the lack of a red-sensitive photoreceptor type present in e.g. humans and other primate species, rodents are often considered functionally blind under red light and vision researchers are regularly using this property in their experimental design. Here, the authors demonstrate that rats preserve their visual capacity in an orientation discrimination task under red light illumination up to ~650 nm, with reduced performance at ~730 nm and no visual capacity for wavelengths of 850 and 930 nm. These results resonate well with another recent study reporting robust photoreceptor responses upon red light application in rats (Niklaus et al., 2020). Together, this challenges the view that red light is invisible to rodents and suggests to reconsider common experimental practices.

In general, this study is well-written, the data is clearly presented and the conclusion that rats maintain their visual capacity under red light is supported by the data. The study addresses an important question in animal research and I believe that the results will be of great interest to the vision research community – especially because rodents have become a prominent model species in vision research over the last decade.

However, I have the following comments:

1. The photon absorption rate depends on the sensitivity curve of the visual pigment expressed by the photoreceptor and on the light intensity. As stated by the authors in the Discussion, it has been shown that mammalian photoreceptors can be activated by infrared light if the intensity is strong enough – both by single photon and two photon events. As a result of this dependence on intensity, it is critical that the authors carefully measure and report the light intensity used in their experiments and ideally relate this to the expected photoreceptor activation, e.g. as photoisomerizations per second. In addition, I suggest to discuss the rationale behind using this intensity range in their experiments. Together, this will help other researchers to interpret these interesting behavioral results and adjust their experimental setup accordingly.

2. With respect to intensity calibration of light sources, I suggest to include additional information in the manuscript. The authors mention in the Methods that they use a calibrated spectrometer for intensity measurements. Does this also account for different sensitivity of the spectrometer across different wavelengths? How do you transform spectrometer output to irradiances shown in Figure 1—figure supplement 2? And what is the unit of irradiance in this figure? Did the authors calibrate the different LEDs such that they result in the same irradiance during experiments? I suggest to include all these points into the Methods section. Also, please use absolute and not normalized spectra to compare intensities across LEDs – see Figure 1D.

3. In addition, I believe it is important to (i) state the absolute intensities used in the experiments in the Results section, (ii) relate these intensities to expected photoreceptor activation and also intensities used in common experimental designs, and (iii) discuss the rationale behind using these intensities during experiments. This will help the reader to interpret the results. For example, did the authors pick these intensities because they are regularly used to dark adapt rats during experiments? Did the authors also test other intensities?

4. Finally, I suggest to add a simple control experiment to exclude any light contamination in the experimental setup. For that, the authors should position the spectrometer head at the position of the rat's head during behavioral experiments and record both background light levels and spectra of all LEDs. While this is unlikely to interfere with the experiments, it is a fast test that could be added as a supplementary figure.

---

## [Author Response]

Essential revisions:1) We recommend that the authors comment on at least three applications that they haven't mentioned, which are additional points of impact of the study:i) Studies of visual development and plasticity in rodents often use "dark-rearing" or days/weeks of darkness as a way to test the visual system under conditions of no stimulation; however, red light is used in husbandry tasks during these times, and any visual experience during those times could severely confound the interpretation of such studies (to pick one recent example, Kowalewski.… Kuhlman Current Biology 2020).ii) Red light sources are often used in optogenetic perturbation experiments and might be assumed to be invisible to the subject, an assumption clearly invalid. In fact there was a paper with a similar finding to the authors (and to Niklaus et al) in this context – Danskin… Waters Plos One 2015.iii) Quite some studies nowadays do eye or whisker monitoring and some conclusions of neural activity in the dark could very well be affected by these conclusions. Another point is place field monitoring in the dark – an experiment that has been quite fundamental for the hippocampal field.

Beyond expanding our citation list of rodent behavioral studies under red or mixed red/infrared light, we have also included discussion of the three applications pointed out by the reviewers. The relevant sections of text are the following:

“The literature includes place-cell studies of rats navigating under dim red light, taken as the absence of visual input (O’Keefe, 1976; Save et al., 2000; Zhang and Manahan-Vaughan, 2015; Zhang et al., 2014). In a developmental study of visual cortex, mice were dark-reared but were allowed very brief daily exposure to red light(Kowalewski et al., 2021).”

“The sensitivity of mice is emphasized by the finding that red illumination delivered into the brain in an optogenetic protocol evoked a behavioral artifact during performance of a visually guided discrimination task (Danskin et al., 2015).”

2) The photon absorption rate depends on the sensitivity curve of the visual pigment expressed by the photoreceptor and on the light intensity. As stated by the authors in the Discussion, it has been shown that mammalian photoreceptors can be activated by infrared light if the intensity is strong enough – both by single photon and two photon events. As a result of this dependence on intensity, it is critical that the authors carefully measure and report the light intensity used in their experiments and ideally relate this to the expected photoreceptor activation, e.g. as photo-isomerizations per second. In addition, we suggest to discuss the rationale behind using this intensity range in their experiments. Together, this will help other researchers to interpret these interesting behavioral results and adjust their experimental setup accordingly.The authors did not attempt to present a matching white light that would activate the rods or the cones in the same manner and investigate adaptation. This should be discussed and considered. Furthermore analysis on the extent to which the cones and the rods would be driven in the red light range would be very helpful.The way in which the LED data is presented and analyzed should be improved. In particular, the authors seem to show fits and they are shown on a linear scale. The authors should show the raw spectrometer values and show these on a logarithmic scale in proper physical units in order to assess how much energy the LED generates in the normal vision range. With respect to intensity calibration of light sources, we suggest to include additional information in the manuscript. The authors mention in the Methods that they use a calibrated spectrometer for intensity measurements. Does this also account for different sensitivity of the spectrometer across different wavelengths? How do you transform spectrometer output to irradiances shown in Figure 1—figure supplement 2? And what is the unit of irradiance in this figure? Did the authors calibrate the different LEDs such that they result in the same irradiance during experiments? We suggest to include all these points into the Methods section. Also, please use absolute and not normalized spectra to compare intensities across LEDs – see Figure 1D. In addition, we believe it is important to (i) state the absolute intensities used in the experiments in the Results section, (ii) relate these intensities to expected photoreceptor activation and also intensities used in common experimental designs, and (iii) discuss the rationale behind using these intensities during experiments. This will help the reader to interpret the results. For example, did the authors pick these intensities because they are regularly used to dark adapt rats during experiments? Did the authors also test other intensities?If there would be any concern left here after these analysis, one solution could be to use monochromatic lasers for the experiment.We further suggest to add a simple control experiment to exclude any light contamination in the experimental setup. For that, the authors should position the spectrometer head at the position of the rat's head during behavioral experiments and record both background light levels and spectra of all LEDs. While this is unlikely to interfere with the experiments, it is a fast test that could be added as a supplementary figure.

We have broken this extended editorial point (2) into a series of problems.

Regarding “discussing the rationale behind using this intensity range in their experiments.” The rationale behind using this intensity range was not to match or mismatch the absorption spectrum of photoreceptors; instead we chose popular, low-cost, commercially available LEDs to mimic the illumination practiced by many laboratories in behavioral neuroscience. However, unlike the uncontrolled “red-tinted” light bulbs used in many studies (examples cited in the Discussion), we chose narrow band LEDs to eliminate the possibility that the tail of a red-centered but wide power spectral distribution might lead to off-peak illumination.

The revised sentence in the manuscript, with new phrasing is:

“In the present work, Long-Evans rats demonstrated substantial visual form capacity under illumination by LEDs emitting long-wavelength red and far-red light – LEDs that are low-cost, easy to use, and thus common in the laboratory.”

Next, we reply to queries about the presentation and analysis of LED data. In the original manuscript we included the relative raw spectral irradiance measures in log-scale in Figure supplement 2. We also included the relative log values (without fitting) in the Figure 1D inset. The values in the log plots were normalized such that the highest light intensity, which belongs to the white LEDs, was 1. Normalization emphasized that the <580 nm-irradiance emitted by the red (626, 652, and 729 nm) LEDs was 2-50 times less than that of the Infrared LEDs, which we know was insufficient to support vision based on the behavioral results. Considering the weak irradiance below 580 nm, visual performance under the red LEDs was nearly impossible to explain by spreading of red LED emission to short-wavelength bands; a more plausible interpretation has to be that rats could see under the longer wavelengths emitted by the red LEDs.

Integrating the power under the curves at wavelengths <580 nm reveals that the infrared LEDs emitted from 2 to 50 times higher intensity as compared to red LEDs, in spite of the infrared LEDs being centered at longer wavelengths (infrared 854 nm and 930 nm LEDs emitted 1.95 and 0.48 mW/cm^2^, respectively, while red 626, 652, and 729 nm LEDs emitted 0.12, 0.28, and 0.04 mW/cm^2^, respectively). Even if the infrared LEDs produced higher intensity of short wavelength light, rats performed at chance under illumination with IR LEDs, providing further evidence that visual performance under red light could not be well explained by unintended “leakage” towards shorter wavelengths. If irradiance below 600 nm were the cause of residual performance under 626, 652, and 729 nm LEDs, then performance would have been better, not worse, under the 854 and 930 nm LEDs.

As recommended by the reviewers, instead of normalized intensity values we have now presented the non-normalized raw spectrometer intensity values in logarithmic scale in units of µW/m^2^/nm in Figure 1D.

In addition, we have included Supplementary file 1 summarizing the intensity of each LED in mW/cm^2^ and total irradiated power (in mW), as measured through our spectrometer (fiber+cosine corrector with a diameter of 3900 µm) from direct measurement of the LEDs, together with the forward current with which the LEDs were supplied and the power rating from the manufacturer datasheet (for further sanity checking as well as replication).

As a control, we can compare total measured irradiated power (the integral of the power spectral density × the surface area of the spectrometer probe and cosine corrector; table column 3) with expected radiated power (table column 6). Considering that optical coupling between the LED and the optical port of the cosine corrector was not perfect (some light escaped laterally, and the optical port was not at exactly zero distance from the surface of the LED) we can verify that the maximum irradiated power for each of our LEDs used in the experiments was in most cases approximately the same as what would be expected from the datasheet (compare columns 3 and 6). An exception was the 729 nm LED which produced less irradiated power than expected from the datasheet. However this discrepancy – which we are not able to explain – does not affect the study’s main conclusions inasmuch as rats showed good visual performance in spite of the low measured irradiance; output around the peak was sufficient to support visual discrimination.

Regarding the photon absorption rate, the typical powers used in microscopy to generate two-photon excitation without damaging the biological samples are between 1 and 10 mW for the mean incident power. For instance for a laser beam of 1 mW, at a rate of 76 MHz with 100 fs pulse duration, and with an objective covering a surface in the focal plane of 1 μm^2^, the peak power is 0.13x10^9^ W/cm^2^ at the focal plane of the sample (Cardin et al., 2010; Lévêque-Fort and Georges, 2005). These intensity values are many orders of magnitude higher than the LEDs in our work. For example, in the case of the 626 nm red LED we have: 19.65x10^-3^ W/cm^2^/0.13x10^9^W/cm^2^ = 1.5115x10^-10^, which means our LED is 10 orders of magnitude less powerful than typically needed to achieve the two-photon quantum IR→UV transition.

Regarding spectrometer calibration, we have included further details in the Methods section. We did not directly calibrate the LED output but rather used a calibrated spectrometer to measure the spectral intensity of each light source. The calibration of the spectrometer precisely accounts for the different sensitivity of the spectrometer across different wavelengths. The spectrometer we used has a software suite that reports the values measured from the calibrated spectrometer as spectral irradiance (the power received by a surface per unit area of the measurement probe per wavelength (in units of microwatts per square centimetre per nanometer [µW/cm^2^/nm^2^]). We have included these additional details in the Methods section as requested by the reviewers.

Regarding the suggestion to present a matching white light that would activate the rods or the cones in the same manner, we note that the IR LED spectrum in the visible range is mostly flat (white). This already provides a matching white light, as noted in a preceding reply: in spite of the higher emission at short wavelength, rats performed at chance under illumination with IR LEDs providing further evidence that visual performance under red light could not be well explained by unintended “leakage” towards shorter wavelengths. If irradiance below 600 nm were the cause of residual performance under 626, 652, and 729 nm LEDs, then performance would have been better, not worse, under the 854 and 930 nm LEDs.

(i) State the absolute intensities used in the experiments in the Results section,

We have done this in Figure 1D and Figure-supplement S2.

(ii) Relate these intensities to expected photoreceptor activation and also intensities used in common experimental designs.

We followed the procedure explained by (Franke et al., 2019). Briefly, for each LED we computed the following measurements which we have included as the supplemental information.

Using the spectrophotometer measurements (P(λ)[μW.cm−2.nm−1]), we can calculate the spectral power density for each LED:

ILED(λ)=10−2×∫λminλmaxP(λ)d(λ)[W.m−2.nm−1] The energy of a single photon at wavelength λ is Qp(λ)=h⋅c/λ[m2.kg.s−1], where c=299,792,458m.s−1 is the speed of light, and h=6.62607004×10−34[J.s] is Planck's constant.

Then we computed the spectral photon count for each LED (SPC(λ)[photons.μm−2.s−1.nm−1]):

SPC(λ)=ILED(λ)Qp(λ)=10−12/(h×c)×ILED(λ)×λ[photons.μm−2.s−1.nm−1] Given the d=50cm distance between the LEDs and the stimulus delivery area, the rat’s dilated pupil area of Apupil=0.0314cm2, the rat’s retinal surface area of Aretina=0.8cm2 and the rat’s rod outer segment area of Arod=4×10−8cm2 (Mayhew and Astle, 1997), we approximated the photon flux density (C) of an isotropic LED radiation pattern at each photoreceptor as: C=2.Apupil.Arod4πd2.Aretina∫λminλmaxSPC(λ)d(λ)[photons.μm−2.s−1] Then, the photon flux per photoreceptor is: Cflux=108. C. Arod[photons.s−1] We included a summary of the findings for each LED in the supplemental information.

(iii) Discuss the rationale behind using these intensities during experiments. This will help the reader to interpret the results. For example, did the authors pick these intensities because they are regularly used to dark adapt rats during experiments? Did the authors also test other intensities?

These are low cost and affordable light sources commonly used in neuroscience labs around the world. We chose narrow band LEDs though, because of the risk of lower wavelengths confounding the results. We did not test other intensities.

Regarding the intensities used in common experimental designs we note that in the majority of behavioral neuroscience reports researchers do not report any quantity (e.g. they only report using a “dim” red light), some use photometric measures (e.g.luminance which is expressed in terms of perceived brightness to the human eye hence less meaningful for other species) and a few report radiometric values for their lighting sources. Typical light intensities used in experiments are in the range 1-300 µW/cm^2^. Our LEDs at a distance of 50 cm from the rats’ head had an intensity of approximately 0.03-5 µW/cm^2^.

The estimated intensity at the rat’s eye level for all of our LEDs was above the behavioral threshold of 6.28 x 10^3^ µW/cm^2^ estimated by (Rosenberger and Ernest, 1971) in scotopic conditions. These values are summarized in Author response table 1 below:

**Author response table 1. resptable1:** 

LED type	Total power density (µW/cm^2^)	Power density received by rat’s eye (µW/cm^2^)	Photon flux density by each photoreceptor (photons/µm^2^/s)
White LED	1.14e+06	2.28e+0	1.19e+05
RED 626 nm	1.12e+04	2.25e-02	1.38e+03
RED 652 nm	3.36e+04	6.72e-02	4.31e+03
RED 729 nm	6.78e+03	1.36e-02	9.75e+02
IR 854 nm	2.34e+05	4.68e-01	3.92e+04
IR 930 nm	7.59e+04	1.52e-01	1.40e+04

In reply to the option of using monochromatic lasers for the experiment, the authors appreciate how this approach would “clean up” the issue of wavelength sensitivity, however it would involve repeating the entire study and would digress from the main question posed by the report: Under typical laboratory red-light conditions, do rats see?

Regarding the “simple control experiment to exclude any light contamination in the experimental setup.… position the spectrometer head at the position of the rat's head during behavioral experiments and record both background light levels and spectra of all LEDs,” this would seem to address whether light was present within the apparatus besides the intended LED illumination. We tested this and reported the complete absence of light when LEDs were turned off. The value was 0 cd/mm^2^ (Konica Minolta LS^-1^00 luminance meter, Tokyo). We refer to the text:

“Behavioral testing was done in a light-proof box, where illumination in the absence of the LED sources was 0 cd/mm^2^ (Konica Minolta LS^-1^00 luminance meter, Tokyo).”

(3) The paper needs to be improved in terms of considering the mechanisms:(i) The discussion on this needs to be improved and extended. These findings likely have a fairly straightforward explanation.

To address this, we have included a simple conjecture:

“Although our study does not include direct measures of photoreceptor sensitivity, the results indicate that mechanisms must exist for converting miniscule quantities of receptor activation into meaningful signals, perhaps through some cross-retina population code.”

ii) It goes beyond the scope of the paper to investigate whether this depends on cones, rods or even two-photon effects (note that this is unlikely to explain the data given the wavelength results that they describe, this may have been a bit misplaced). The most straightforward explanation is that the small drive of the cones and rods, combined with adaptation mechanisms, is sufficient to perform the task at hand. This seems testable by using very low-intensity green light (hitting the M-cones and the rods) that elicits a similar drive. This would show there is nothing special about red here, it just means that rats can do this kind of task under very low-level lighting conditions.

The authors appreciate that the present results point to a number of future directions, including the suggested experiment of determining exactly how small an intensity of green light is sufficient to support behavior. While we will entertain this and related problems in the future, the present aim is very simply to ask whether under typical dim red light conditions, rats are deprived of form vision, as is commonly believed. To maintain this as a brief and timely report (that is, without adding the years necessary to retrain and test rats under different conditions), our preference is to make the study tightly bound to the original problem.

(iii) Another very simple experiment that could have been considered is to prevent adaptation, so first prevent adaptation by using green or white light at normal intensity and then put the animal in the red condition – this should prevent the animal from doing the task. The authors write that "Rats were dark-adapted in a light free environment for 20-30 minutes prior to each session". Was this a necessary step?

The adaptation protocol was 20-30 minutes presence in the light-sealed booth, as stated in Methods. The suggested protocol of exposing rats to non-red light and then testing under red light is brilliant in principle. However, it is not clear if the results would advance us much, due to ambiguity. There are two scenarios that might underlie conserved vision under red light.

First, blue-green cones may absorb enough red light to support form perception. In that case, pre-exposure to blue-green light would adapt the cones and they would be less excitable by red light, causing a performance drop.

Second, rods (not cones) may absorb faint red light. But they, too, would be adapted by pre-exposure to blue-green light and their reduced sensitivity would likely cause a drop in performance.

In short, pre-exposure to blue-green would likely reduce the sensitivity of both channels which might support form vision under red light after adaptation to complete darkness. The expected drop in performance would be a neat result but would not take us closer to understanding the basis of form vision under red light.

4) The normalized irradiance plot in 1D does not extend below 350nm though it is stated that rats have cones with peak sensitivity ~360nm. So, is it possible that the 630-730nm wavelength LEDs also give off a small amount of 300-350nm UV light, and this is used by the rats for discrimination? How can this be ruled out? Many long-pass optical filters also pass some light of much shorter wavelengths than the cutoff - to pick one example, see the spectrum for the DMLP650 on this page (https://www.thorlabs.com/newgrouppage9.cfm?objectgroup_id=3313). I don't know whether the LEDs in this study obtain their wavelength selectivity from anything at all related to the mechanisms of filtering in DMLP650 or similar - I'm not an expert - but it seems like it must be ruled out.

Our spectrometer was calibrated in the visible+infrared range, hence the measured wavelengths start from 350 nm. Regarding the emission mechanism, the LEDs used in our work were InGaAlP (Aluminium gallium indium phosphide). These alloys are usually used to make red, orange and yellow LEDs and they do not emit in the blue and UV range because their emission band is limited to the bandgap of the alloy material which is very finely tuned to a specific wavelength with very narrow half-width of the output (~10-20 nm). Assuming a Gaussian output profile (from our measurements as well as the manufacturer’s data sheet) the power at 300-350 nm is smaller than the peak by roughly 10^-40^. In addition we note again the behavioral evidence previously mentioned in reply to the major concerns regarding short wavelengths as well as the matching white light.

5) What accounts for the difference in findings relative to Jacobs et al 1991? I think this paper is the most classic/foundational, and it included a behavioral test of visual function, but concluded no behavioral sensitivity beyond 600 nm. I think the authors ought to indicate what they think accounts for the discrepancy in conclusions between that work and the present. Difference in species? Difference in behavioral task design? Difference in light source? Difference in range of wavelengths tested? Different analytical methods used to draw conclusions?

1) One possibility is that the spectral sensitivity curves estimated by Jacobs were not accurate for wavelengths longer than 600nm. A more recent study (Rocha et al., 2016) using a more sensitive technique called AC Constant-Response Method expanded the available data points to beyond 600nm.

2) Flicker stimuli used by studies such as Jacobs and Rocha could be less sensitive in determining the retinal absorption of longer wavelengths. Niklaus et al., 2020 argue that their single flash method was substantially more acute than the constant flicker method in detecting responses in the low sensitivity range. Furthermore, retinal responses to flicker might not map directly to form vision under adapted conditions.

3) Regarding the behavioral experiments, Jacobs et al simply tested color (spectral sensitivity) discrimination. In other words, they did not (and could not) determine the ability of their test subjects to carry out visual “form” discrimination. The behavior was not comparable to the visual object discrimination, in real world conditions, in our task. In fact, we do not claim that rats can distinguish red from other colors. We simply state that rats can see under red monochromatic illumination, most likely based on a rod-based (together with M cones) mechanism. Additionally, Jacobs et al, write that "Ceiling-mounted fluorescent tubes were used to illuminate the test chamber (ambient illuminance 70 lx)" while testing the animals. This is clearly a photopic task designed for testing color discrimination. In short, Jacobs et al did not (and could not) conclude “no behavioral sensitivity beyond 600 nm.”

4) Moreover, the behavioral sensitivity of rats was a function of background lighting as shown in Jacobs et al., 2001. The results show spectral sensitivity functions for four different achromatic backgrounds [from top to bottom: log(luminance=1.12, 1.43, 1.76 and 2.09, where luminance is measured in cd.m^−2^]. Therefore in scotopic conditions one might expect higher sensitivity of photoreceptors to longer wavelengths.

5) In a relevant study, (Williams et al., 2005) trained wild-type as well as coneless mutant mice to discriminate a 600nm test light on an achromatic background (-1.4 log scot. cd/m^2^). At intensities around 10^13^ photons/s/cm^2^ mice achieved ~70% discrimination performance. All red light sources used in our work emitted approximately 10^19^ - 10^20^ photons/s/cm^2^ at rat’s cornea level. Williams et al., 2005 shows psychometric functions for wild-type mice (squares) and coneless mice (triangles) derived for detection of two monochromatic test lights (~500nm and 600nm) added to a dim (-1.4 log scot. cd/m^2^) background. Importantly in this study the authors could infer the nature of the photopigment underlying this behavior and concluded that at low-level light adaptation both coneless and wild-type mice have equally effective rod-based vision.

6) There might be significant species related sensitivity differences. For instance even among different strains of rats, (Niklaus et al., 2020) showed lower scotopic a- and b-wave amplitudes in red and white ERGs in albino Winstars compared to the pigmented Brown Norway rats.

7) Figure 3 of Jacobs et al 1991 shows 1-2 orders of magnitude loss of short-wavelength sensitivity consequent to long-wavelength adaptation. This suggests that the long-wavelength receptors, now adapted, were previously contributing to the short-wavelength response.

6) In the introduction, it is stated that Rocha et al 2016 found no ERG response above 620nm, while in the discussion it is stated that Niklaus et al 2020 found ERG responses to 650nm. Are the authors claiming that Rocha et al 2016 are simply wrong? If so, it would seem more appropriate to raise that study and call it into question in the discussion rather than the introduction - as is, that study is presented as accepted fact and is never revisited or questioned, which I found very confusing.

There are several experimental differences between the two cited studies, as pointed out by (Niklaus et al., 2020). Importantly Rocha et al (2016) examined spectral sensitivities under light adaptation, which leads to degradation of response to long wavelengths. The light adaptation procedures performed by the authors aimed to measure cone responses without rod influence.

Another difference is that (Niklaus et al., 2020) used a potentially more sensitive technique based on single flash ERG in the dark- as well as light-adapted retina, whereas Rocha et al. (2016) used flicker stimuli between 4 and 12 Hz light on light-adapted retina.

Beyond these important differences in methods, upon a more careful examination of (Rocha et al., 2016) results, as well as their supplemental data, we found their statement regarding retinal sensitivity to long wavelengths not well supported by their own data: there was in fact an ERG response above 620nm.

This is also evident from the poor fit of the measured data as shown in figure 3B-D of Rocha et al., 2016.